# High-Pressure Synthesis of Cubic ZnO and Its Solid Solutions with MgO Doped with Li, Na, and K

**DOI:** 10.3390/ma16155341

**Published:** 2023-07-29

**Authors:** Nikolai O. Taibarei, Vladimir G. Kytin, Elizaveta A. Konstantinova, Vladimir A. Kulbachinskii, Serguei V. Savilov, Vladimir A. Mukhanov, Vladimir L. Solozhenko, Vadim V. Brazhkin, Andrei N. Baranov

**Affiliations:** 1Department of Chemistry, Lomonosov Moscow State University, 119991 Moscow, Russia; renoa@inbox.ru (N.O.T.); savilov@chem.msu.ru (S.V.S.); v.mukhanov1@gmail.com (V.A.M.); 2Department of Physics, Lomonosov Moscow State University, 119991 Moscow, Russia; kytin@mig.phys.msu.ru (V.G.K.); konstantinova@physics.msu.ru (E.A.K.); kulb@mig.phys.msu.ru (V.A.K.); 3LSPM-CNRS, Université Sorbonne Paris Nord, 93430 Villetaneuse, France; vladimir.solozhenko@univ-paris13.fr; 4Institute for High Pressure Physics, Russian Academy of Sciences, Troitsk, 108840 Moscow, Russia; brazhkin@hppi.troitsk.ru

**Keywords:** zinc oxide, magnesium oxide, rock-salt structure, doping, group I metals, high pressure

## Abstract

The possibility of doping ZnO in its metastable rock salt structure with Li, Na, and K intended to act as acceptor dopants was investigated. For the first time, Mg_x_Zn_1−x_O alloys and pure ZnO with a rock salt structure doped with Li, Na, and K metals was obtained by high-pressure synthesis from pure oxides with the addition of carbonates or acetates of the corresponding metals as dopant sources. Successful stabilization of the metastable rock salt structure and phase purity were confirmed by X-ray diffraction. Transmission electron microscopy was used to study the particle size of nanocrystalline precursors, while the presence of Li, Na, and K metals in rock salt ZnO was detected by electron energy-loss spectroscopy and X-ray photoelectron spectroscopy in Mg_x_Zn_1−x_O alloys. Electron paramagnetic resonance measurements revealed the acceptor behavior of Li, Na, and K dopants based on the influence of the latter on native defects and natural impurities in ZnO-MgO alloys. In addition, diffuse reflectance spectroscopy was used to derive band gaps of quenched rock salt ZnO and its alloys with MgO.

## 1. Introduction

Zinc oxide has long been considered as a promising semiconductor material for the fabrication of transparent bipolar electronic devices [1]. ZnO has a wide band gap of 3.37 eV [2] and, like most other wide-band-gap semiconductors, exhibits a strong tendency toward *n*-type conductivity [3], with the exception of GaN [4]. While ZnO has distinct advantages over GaN, namely, its greater accessibility due to natural abundance as well as a much higher exciton binding energy (60 meV [5] vs. 28 meV [6]), a stable and reproducible *p*-type ZnO remains elusive, which currently limits the realization of ZnO-based electronic technology.

In the last two decades, a remarkable amount of research has been devoted to achieving *p*-type conductivity in ZnO by doping with various elements; the results, however, have been controversial. These works can be mainly divided into two groups: doping with group V elements (N, P, As, Sb) as oxygen substitutes [7,8,9,10,11] and doping with group I elements (Li, Na, K) as Zn substitutes [12,13,14]. Doping with group V elements typically leads to the formation of deep acceptor levels or is compensated by various self-compensation mechanisms, such as the formation of Zn or hydrogen interstitials or oxygen vacancies [15]. Substitution of Zn with group I elements in ZnO, which crystallizes in a thermodynamically stable hexagonal wurtzite structure (w-ZnO), is difficult because these elements primarily enter interstitials and act as donors rather than substituting Zn in the lattice [16].

Therefore, an interesting way to overcome the above challenges would be to take a step away from the wurtzite structure. The only other reliably known crystal structure of ZnO is cubic or rock salt (rs-ZnO), which is metastable and exists at pressures above a certain threshold [17,18]. However, it is possible to stabilize bulk rs-ZnO at ambient conditions either by alloying with appropriate amounts of other oxides with a cubic structure, i.e., MgO, above the pressure of the wurtzite-to-rock-salt transition threshold [19,20,21,22] or by converting nanocrystalline powders of w-ZnO at high pressure [23]. In addition, rs-ZnO can be stabilized in thin films via epitaxy using a cubic substrate, bringing the advantages of thin film technology to this potential semiconductor material [24].

Recently, it has been shown by first-principles calculations [25] that doping rs-ZnO with Li can lead to creation of shallow acceptor levels with predicted hole concentrations exceeding 10^19^ cm^−3^ while suppressing the formation of undesired self-compensating defects due to the shift of the valence band closer to vacuum compared to the ground-state w-ZnO. However, to date, there are no experimental attempts to obtain rs-ZnO or its solid solutions doped with Li or other alkali metals. In this work, we provide and characterize an experimental approach for the synthesis of rock salt Mg_x_Zn_1−x_O alloys as well as stabilized nanocrystalline rs-ZnO doped with group I elements (namely, Li, Na, and K) and investigate the influence of these metals on the acceptor behavior.

## 2. Materials and Methods

In the present work, two different systems with the rs-ZnO structure were studied: Mg_x_Zn_1−x_O alloys doped with (Li, Na, and K) metals, where the nature of the dopant as well as MgO content were varied, and rs-ZnO doped with the same metals.

### 2.1. Synthesis of Mg_x_Zn_1−x_O Alloys Doped with Li, Na, and K Metals

The first group of metal-doped Mg_x_Zn_1−x_O alloys (denoted as MZnn:X, where “nn” is the amount of MgO in mol% and X represents Li and alkali metal introduced into the sample or is absent if no dopant cation was introduced) were synthesized from suitable amounts of ZnO and MgO powder of high purity (preliminarily annealed at 400 °C for 1 h to remove residual moisture), which were mixed, thoroughly ground, and then milled on a ball mill several times in order to improve homogeneity by reducing the average particle size. The next step was designed to introduce Li, Na, and K metals into the precursors. Prepared ZnO-MgO mixtures were additionally ground and milled with anhydrous carbonates of the respective dopant metals taken in the amount necessary to introduce 5% of dopant cation X^+^ in relation to total amount of Zn^2+^ and Mg^2+^. The obtained precursors were annealed for 8 h in a tube furnace at 900 °C in a dynamic vacuum of 2–3 mbar so to ensure thermal decomposition of the carbonates. Note that Li_2_CO_3_ decomposes slightly above its melting point (720 °C), while Na_2_CO_3_ and K_2_CO_3_ melt without decomposition (~850 and ~900 °C, respectively) at atmospheric pressure. However, Na_2_CO_3_ and K_2_CO_3_ start to decompose slowly right after reaching their melting point, which is why vacuum and long annealing are required [26,27]. The process can be summarized by the following schematic equation:(1)$${\mathrm{Mg}}_{x}{\mathrm{Zn}}_{1-x}O+\frac{y}{2}X_{2}{\mathrm{CO}}_{3}\to {\mathrm{Mg}}_{x}{\mathrm{Zn}}_{1-x}X_{y}O_{1+\frac{y}{2}}+\frac{y}{2}{\mathrm{CO}}_{2}↑$$
where y = 1:19.

High-pressure synthesis was performed in a toroid-type high-pressure apparatus using a pyrophyllite cell with a graphite heater operated by direct current, as described elsewhere [28]. Precursor powders were encapsulated in gold foil to reduce sample contamination with carbon by active diffusion of the latter. Acquired precursors were compressed at 4 GPa in the 800–1000 °C temperature range and then quenched. It was found that any temperature within the above range is suitable for a successful wurtzite-to-rock-salt transition and does not lead to significant changes in the subsequent properties of the samples. Therefore, a temperature of 800 °C was preferred because higher temperatures induce more carbon diffusion into the sample.

### 2.2. Synthesis of rs-ZnO Doped with Li, Na, and K Metals

The absence of MgO, which naturally crystallizes in the cubic structure, requires an alternative approach to stabilize otherwise metastable rs-ZnO. This could be achieved by high-pressure treating of nanocrystalline w-ZnO with a narrow particle size distribution [23], prepared by precipitation of w-ZnO from alcohol solutions of Zn(CH_3_COO)_2_. However, introduction of a controlled amount of an alkali cation is not possible in this case. Therefore, an original technique for the synthesis of dopant metal containing fine w-ZnO particles was developed. Aqueous solutions of Zn(CH_3_COO)_2_ and CH_3_COOX (where X = Li, Na, or K) in molar ratio of 19:1 were pulverized in liquid nitrogen and freeze-dried. The resulting dry product was milled in the presence of a salt matrix at a ratio of 1 to 10. Anhydrous carbonates of the same alkali metal were used as the salt matrix for each respective sample in order to facilitate subsequent acetate decomposition. The milled powders were annealed at 550 °C for 24 h, after which the salt matrix was removed by dissolution in water. Precipitates of w-ZnO nanoparticles were separated from aqueous solutions by centrifugation, washed with ethanol, and dried at 90 °C for 3 h. The obtained precursors were encapsulated in gold foil and converted to rs-ZnO at 7.7 GPa and ~600 °C. The samples were designated as pure rs-ZnO or rs-ZnO:X for samples doped with Li, Na, and K metals.

X-ray diffraction (XRD) patterns were collected on a Rigaku D/MAX-2500V/PC X-ray diffractometer with rotating anode and a 1D scintillation detector (CuKα1 radiation, λ = 0.15418 nm; scan step size of 0.02°; 2θ range 20–80°). Transmission electron microscopy (TEM) study of precursors and rs-ZnO nanoparticles was performed on a JEOL JEM-2100, Akishima, Japan microscope equipped with an electron energy-loss spectroscopy (EELS) attachment. X-ray photoelectron spectra (XPS) were acquired on an Axis Ultra DLD spectrometer (Kratos Analytical, Stretford, UK) with a monochromatic AlKα source (E = 1486.7 eV). Charge neutralization was applied, resulting in the C1s peak position of 284.8 eV. The analyzer pass energies were 160 eV for survey spectra and 40 eV for high-resolution scans. XPS spectra were fitted by Gaussian convolution functions with simultaneous optimization of the background parameters. Diffuse reflectance (DR) spectra were recorded using a Lambda 950 spectrophotometer (Perkin Elmer, Waltham, MA, USA). The recovered samples were ground into powders and placed in a special aluminum cell with a quartz window designed to form a dense uniform powder layer. Electron paramagnetic resonance (EPR) spectra were recorded at room temperature by a ELEXSYS-500 EPR spectrometer (Bruker, Berlin, Germany) (X-band, sensitivity is around ~10^10^ spin/g).

## 3. Results

The Rs-Mg_x_Zn_1−x_O system was studied in the concentrations range of 0.1 < x < 0.3. According to XRD, all samples with 30 mol% of MgO yielded a pure rock salt phase after high-pressure treatment (Figure 1). Samples with 20 mol% of MgO generally converted to the cubic structure, but sometimes a small amount of w-ZnO could be detected. Samples with 15 mol% of MgO or less completely converted back to separate phases of cubic MgO and w-ZnO after pressure release. It is important to note that w-ZnO lines are broad in this case, unlike those of the initial ZnO-MgO powder, indicating a decrease in the degree of crystallinity caused by high-pressure treatment, which, in turn, implies that conversion to the rock salt phase occurred under the synthetic conditions, but 15 mol% of MgO or less is not sufficient to stabilize such an alloy at ambient conditions. Consequently, the threshold concentration of MgO required to stabilize a rs-Mg_x_Zn_1−x_O phase is in the range of 0.15 < x < 0.2.

The samples recovered from the high-pressure cell were dense bulks ranging in color from white to gray color. According to two-probe measurements, the bulks had no significant conductivity; consequently, it was not possible to perform Hall measurements and reliably determine the conductivity type and carrier concentrations.

The mean size of crystalline domains *d* for w-ZnO in MZ15, estimated from the Scherrer Equation (2), where *K* is the shape factor (equal to 0.9), *λ* is the X-ray wavelength, *β* is the line broadening (FWHM), and *θ* is the Bragg angle, is about 17 nm compared to 35 nm in the initial powder. At the same time, the size of MgO crystallites increases after the reverse transition, as indicated by the narrowing of the lines. However, it is not possible to evaluate the mean size *d* in this case, because the lines are clearly a superposition of at least two cubic (Fm-3m) phases, which is evident from the double maxima.
(2)$$d=\frac{K\lambda }{\beta cos\theta }$$

Based on the above considerations, Li, Na, and *K* doped ZnO-MgO systems were studied primarily for two different MgO concentrations: 20 and 30 mol%. Lower MgO concentrations are not sufficient to stabilize the rock salt structure, and introduction of dopant cations, even though they prefer octahedral coordination, does not provide a sufficient stabilizing effect. At the same time, higher MgO concentrations would lead to an excessive widening of the band gap *E_g_* if a homogeneous phase was formed. rs-Mg_x_Zn_1−x_O phases are expected to obey Vegard’s law (3) over the entire stability range, where the bowing parameter *b* would be very small due to a minimal mismatch between Zn^2+^ and Mg^2+^in octahedral coordination, making the *E_g_*(*x*) dependence nearly linear [29].
(3)$$E_{g}^{{\mathrm{Mg}}_{x}{\mathrm{Zn}}_{1-x}O}\left(x\right)=xE_{x}^{\mathrm{MgO}}+\left(1-x\right)E_{g}^{\mathrm{ZnO}}-bx\left(1-x\right)$$

After vacuum annealing, no lines other than MgO and w-ZnO were found in the MgO-ZnO:X precursors, indicating complete decomposition of the carbonates (for XRD data, see Appendix A). High-pressure treatment leads to a complete transformation of the precursors into a rock salt phase, in which a small amount of a wurtzite phase can be detected in samples containing 20 mol% of MgO. Note that the highest intensity (101) line of w-ZnO overlaps with the (111) line of the cubic phase, so they can interfere with each other depending on which phase is dominant.

Nanocrystalline precursors doped with Li, Na, and K metals for the synthesis of rs-ZnO obtained by the freeze-drying technique exhibit significantly larger particle size (some of them being larger than 100 nm in diameter), with a rather wide size distribution compared to the previously reported [23] precipitation of nanosized ZnO from alcohol solutions, as can be seen from TEM images (Figure 2a). Electron diffraction of all samples is consistent with the wurtzite structure of the nanoparticles, and the peak around 1080 eV in the energy-loss spectrum confirms the presence of Na in the nanoparticles.

Despite the larger average particle size, there were no micron-sized particles observed that could become areas of spontaneous w-ZnO nucleation; consequently, high-pressure processing still leads to an almost complete conversion of such w-ZnO into the rock salt structure, with only a small amount of the wurtzite phase present, according to XRD (Figure 3). This tendency holds not only in the case of Li, Na, and K doped rs-ZnO, where X^+^ cations could be an additional stabilizing factor due to their preference for octahedral coordination, but also in the case of undoped ZnO obtained by the same synthetic procedure. The mean crystallite size estimated from Equation (2) is about 35 ± 5 nm for all samples and does not change after the transition to the rock salt structure, suggesting that the particles seen in the TEM images of the precursors (Figure 2a) are likely to be individual crystallites, while the particles of rs-ZnO (Figure 2b) are clearly aggregates formed due to partial sintering in the high-pressure cell. The calculated average crystallite size is slightly lower compared to the limit 45 nm estimated in [23] necessary to suppress spontaneous intragrain nucleation, which explains successful stabilization despite the larger average particle size of the precursors. Also, the mean crystallite size is universal across all samples, indicating that a low concentration of dopant cation does not significantly affect the phase transition (for other TEM images, see Appendix A). Electron diffraction (Figure 2, left inset) is in agreement with XRD data and confirms the formation of a rock salt structure.

Despite the successful stabilization of rs-ZnO synthesized by freeze-drying technique, obtained bulks were fragile and crumbled into small pieces after recovery. In addition, as previously described [23], rs-ZnO stabilized by nanocrystallinity exhibits poor thermal stability and undergoes almost complete rock-salt-to-wurtzite transition above 100 °C. All of these factors made it impossible to properly perform Hall measurements on our samples, but simple two-probe measurements showed that rs-ZnO does indeed possess noticeable conductivity. Overall, even the samples preserved in an inert atmosphere still converted back to w-ZnO over the course of the next two years.

In principle, there are two types of positions in a rock salt structure that an alkali cation can occupy: octahedral sites and tetrahedral interstitials. When substituting an X^2+^ cation that occupies an octahedral position, the Li, Na, and K dopant is expected to act as an acceptor, and when positioned in an interstitial, it would act as a donor. In general, dopant cations prefer octahedral coordination, but this may be altered by high-pressure treatment followed by rapid quenching. High-resolution XPS spectra of the elements Li, Na, and K were recorded for all MZ20:X and MZ30:X samples (for spectra overview, see Appendix A). For all lines (Figure 4) except for MZ20:Na, a satisfactory fit can be achieved with only one Gaussian component (in the case of K2p there is spin-orbit splitting into K2p_3/2_ and K2p_1/2_ components), implying that there is essentially only one type of dopant atom. While there are no precise data on the binding energies of Li, Na, and K incorporated into ZnO or MgO, it is most likely that the observed lines correspond to lattice substitution (which would be an equivalent of M^+^ oxidation state) simply because it is extremely unlikely that all dopant atoms occupy only interstitials (M^0^ oxidation state). There is also a distinct additional component of the Na1s peak in MZ20:Na at 1072.6 eV. This component is probably due to Na precipitated on the crystallites surface, especially in the form of carbonates or hydrocarbonates formed by exposure to atmosphere. Deconvolution of the C1s peak confirms the presence of a component at 288.9 eV, usually corresponding to CO_3_^2−^ bound carbon.

It appears that the binding energy is sensitive to the amount of MgO in the alloy, which directly affects the cell parameter *a* of the Fm-3m structure. This effect is particularly noticeable in Li-doped samples (Figure 4a,d), where the binding energy shifts from 55.3 eV for 20 mol% of MgO to 54.6 eV for 30 mol% of MgO. This tendency is weaker in the case of Na doping, where the shift of the Na1s peak is only ~0.4 eV (Figure 4b,e), and almost nonexistent in the case of K doping (Figure 4c,e). Such an effect could be due to a change in the chemical environment caused, for example, by a core-shell-like distribution of MgO and ZnO [20]. Such uneven distribution of cations would affect mobile Li much more than larger Na and K if the dopant has a preference to incorporate into Mg-rich or Mg-poor regions.

EPR spectra of rs-Mg_x_Zn_1−x_O alloys with 20% and 30% MgO are shown in Figure 5 and Figure 6, respectively. All spectra contain six lines of Mn^2+^ ions, which are commonly found as a natural impurity in MgO. Interestingly, we found that this natural impurity, which is often used as a standard in EPR spectroscopy, although not related to the dopants, can serve as an indicator of the influence of dopant metals on intrinsic defects in rs-Mg_x_Zn_1−x_O.

First, the observed EPR lines of Mn^2+^ are not symmetric. Such a shape suggests that the observed lines are superpositions of lines with slightly different g-factors. This could happen if different Mn^2+^ ions are surrounded by different amounts of Zn^2+^ and Mg^2+^ neighboring ions or different electrically active defects, such as vacancies. Concentrations of Mn^2+^ ions estimated from EPR spectra for all cubic alloys are provided in Table 1.

The effect of Li, Na, and K elements on concentration of the Mn^2+^ ions depends on the MgO content. In the samples with 30% MgO concentration, doping with any dopant metal decreases the Mn^2+^ concentration. It seems that the higher the atomic number of the doping element, the lower the concentration decrease. In the samples with 20% MgO doped with Li, Na, and K metals, concentration of Mn^2+^ is higher than in the undoped sample. However, in the series Li-Na-K for doped samples with 20% MgO, the same tendency occurs: the higher the atomic number of the dopant element, the higher the Mn^2+^ concentration.

An additional EPR line with a g-factor close to two was detected in all samples except MZ30:Li and MZ30:Na. The shape of this line is shown in high-resolution spectra in Figure 7 and Figure 8.

In In the undoped sample MZ30, the line corresponds to a nearly isotropic paramagnetic center with g-factor close to 2.0021. In MZ20:Na and MZ20:K samples the shape of the line is more complicated. The g-factor of the line is close to 2.0013. This line probably originates from bulk oxygen vacancies V_O_^+^ (where V_O_^+^ stands for a single-charged oxygen vacancy) in MgO [30,31]. The EPR line in the undoped sample MZ20 and in the doped sample MZ30:K most likely originates from the surface V_O_^+^-H complex [32,33]. Therefore, we associate the EPR signal with a g-factor close to two with bulk and surface oxygen vacancies that have captured one electron. The total concentrations of bulk and surface V_O_^+^ centers estimated from the EPR signal are shown in Table 2.

As can be seen from Table 2, doping with Li, Na, and K metals significantly reduces the concentration of V_O_^+^ and V_O_^+^-H centers. This can be explained by oxidation of V_O_^+^ centers to nonparamagnetic V_O_^2+^.(double-charged oxygen vacancy) and indicates acceptor behavior of dopant impurities. More effective reduction of V_O_^+^ and V_O_^+^-H center concentration in the samples with 30% Mg content after doping with Li, Na, and K atoms may be due to lower energy of acceptor ionization in the samples with higher MgO content. Decrease of Mn^2+^ concentration in the 30% MgO doped alloys can be explained by oxidation of Mn^2+^ ions to nonparamagnetic Mn^3+^ ions due to acceptor behavior of group I atoms. First-principles calculations predicted that in MgO there is a range of the Fermi level positions where the formation energies of Mn^3+^ and Mn^2+^ are nearly equal [34]. In the presence of oxygen vacancies, the Fermi level could be shifted beyond this range in undoped samples with 30% MgO content. The formation of paramagnetic Mn^2+^ ions is then favored in these samples. Doping with group I atoms shifts the Fermi level down and makes the formation of nonparamagnetic Mn^3+^ ions more favorable. This interpretation suggests that the Fermi energy shift increases in the row K-Na-Li. This means that the ionization energy of the acceptor centers associated with dopant metals decreases as the atomic number decreases. XPS data imply that samples with 20% MgO content most likely consist of Mg-rich and Mg-poor crystallites. Formation energies of Mn ions in Mg-poor crystallites may be different from those in MgO. This could explain why doping of the 20% MgO samples with Li, Na, and K metals does not lead to oxidation of Mn^2+^ ions to Mn^3+^ ions.

Overall, the EPR data indicate the acceptor behavior of dopant metals in rs-Mg_x_Zn_1−x_O alloys. The ionization energy of the corresponding acceptors decreases with the decrease of the atomic number of the group I metal.

Alloying with MgO is a typical way to broaden the band gap in ZnO. To compare *E_g_* of rs-ZnO with rs-Mg_x_Zn_1−x_O alloys, Tauc plots (Figure 9) were obtained from diffuse reflectance spectra using the well-known Equation (4) where the reemission function *F*(*R*) is calculated according to the Kubelka–Munk formula (5).
(4)$$F\left(R\right)h\nu =C{\left(h\nu -E_{g}\right)}^{n}$$
(5)$$F\left(R\right)=\frac{{\left(1-R\right)}^{2}}{2R}$$
where *h* is the Planck constant, ν is frequency, *R* is reflection, and *n* is a parameter that depends on the type of electronic transition where *n* = 1/2 for direct allowed transition, *n* = 3/2 for direct forbidden transition, *n* = 2 for indirect allowed transition, and *n* = 3 for indirect forbidden transition. Rs-ZnO is an indirect band gap semiconductor, so *n* = 2 was used.

One would expect *E_g_* of MZ20 and MZ30 to be approximately 3.51 and 4.05 eV, respectively, assuming the previously reported *E_g_* = 2.45 eV for pure rs-ZnO [35], *E_g_* = 7.77 eV for MgO [36] and *b* = 0 in Equation (3). However, there is almost no difference between the observed band gaps of pure rs-ZnO (2.71 eV), MZ20 (2.57 eV), and MZ30 (2.71 eV), which essentially means that ZnO-MgO alloys have the same absorption edge as pure rs-ZnO. Each spectrum also has a vague “step” at ca. 4 eV, which becomes steeper when reconstructed using *n* = ½ in Equation (4). This additional absorption edge could thus be attributed to a direct transition, which in the case of pure rs-ZnO probably corresponds to the w-ZnO impurity. The obtained *E_g_* = 3.21 eV is somewhat lower than the commonly reported 3.37 eV, but close to that (3.25 eV) recorded for the initial w-ZnO used in the experiments. It is also in good agreement with the experimentally observed E_g_ in similar measurements [37].

MZ20 and MZ30 have direct *E_g_* of 3.54 and 3.62 eV, respectively. Given the absence of the wurtzite phase, this edge most likely corresponds to a rs-Mg_x_Zn_1−x_O phase. Note that MgO itself has a direct band gap. The absolute difference of 0.08 eV should be considered negligible in this case for two reasons. First, when studying mixed phases using diffuse reflectance spectroscopy, it is important to consider additional effects, such as phase dilution [37], which could distort the derived *E_g_* values. This is not possible here due to the unknown stoichiometry and individual band gaps of the mixed phases. Second, it is always somewhat speculative how to determine the linear part of the Tauc plot, even if it returns a very good fit in terms of standard deviation (in our case R_s_ > 0.999 for all linear fits). However, it is clear that rs-Mg_x_Zn_1−x_O alloys are composed of more than one phase with different optical properties, despite their apparent homogeneity according to the visual analysis of the XRD data. For example, a similar situation has recently been demonstrated for related rs-Ni_x_Zn_1−x_O solid solutions which show distinct structural features in the ZnO-enriched region [38,39]. There, nonlinear behavior of thermal expansion coefficients is evident and may indicate that the solid solution in question is nonequilibrium.

In the particular case of the ZnO-MgO system, the Mg-rich phase may have an identical MgO content between MZ20 and MZ30 samples, as indicated by the obtained *E_g_* values for MZ20 and MZ30. Therefore, a much more subtle analysis is required in the future for a better understanding of the actual phase distribution in the ZnO-MgO system.

## 4. Conclusions

Approaches for the high-pressure synthesis of both rs-Mg_x_Zn_1−x_O alloys and pure nanocrystalline rs-ZnO doped with Li, Na, and K have been successfully developed. The estimated concentration of MgO required to stabilize bulk rs-Mg_x_Zn_1−x_O doped with alkali metals was found to be ~20 mol%. Substitution of Zn^2+^/Mg^2+^ cations by Li, Na, and K metals in the cubic structure was suggested by XPS, and the presence of dopant cations inside rs-ZnO nanoparticles was demonstrated by EELS. EPR study reveals that Li, Na, and K metals do indeed act as acceptors in rs-Mg_x_Zn_1−x_O, which is evident from the decrease in the paramagnetic center (V_O_^+^ and V_O_^+^-H) concentrations. Moreover, the effect becomes stronger in the row K-Na-Li, probably due to the decrease of ionization energy of related acceptor centers in the same row. However, the overall effect of doping is rather minimal and seems to be complicated by the tendency of ZnO-MgO alloys to form Mg-rich and Mg-poor phases despite the macroscopically homogeneous nature of the alloys as inferred from the XRD data. This tendency is also strongly suggested by the presence of two absorption edges in the diffuse reflectance spectra, one belonging to an indirect transition in almost pure rs-ZnO and the other belonging to a direct transition in rs-Mg_x_Zn_1−x_O and being almost independent of MgO concentration in the range studied.

## Figures and Tables

**Figure 1 materials-16-05341-f001:**
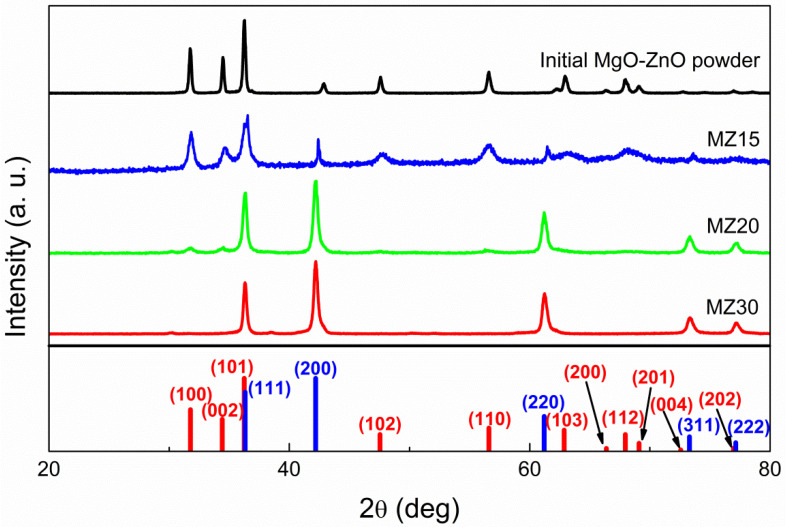
X-ray diffraction patterns of undoped MZ30, MZ20, and MZ15 samples and initial MgO-ZnO mixture. Red bars indicate w-ZnO phase peaks and blue bars indicate rock salt ZnO or Mg_x_Zn_1−x_O phase.

**Figure 2 materials-16-05341-f002:**
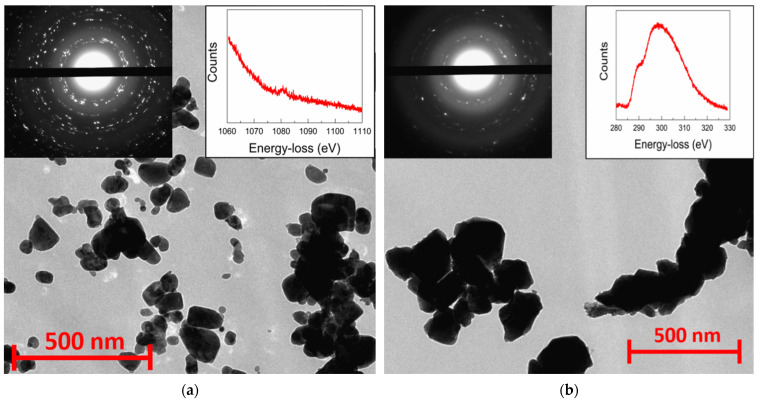
TEM images of w-ZnO:Na nanocrystalline precursor (**a**) and rs-ZnO:K after high-pressure treatment (**b**). Left insets show electron diffraction from the observed area and right insets show energy-loss spectra of Na K-edge (**a**) and K L_2_- and L_3_-edge (**b**).

**Figure 3 materials-16-05341-f003:**
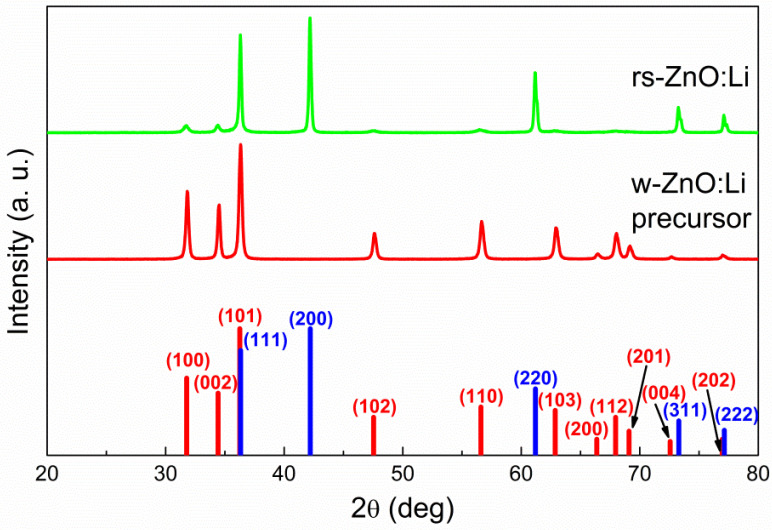
X-ray diffraction patterns of nanocrystalline ZnO before and after high-pressure treatment. Red bars indicate w-ZnO phase peaks and blue bars indicate rs-ZnO phase peaks.

**Figure 4 materials-16-05341-f004:**
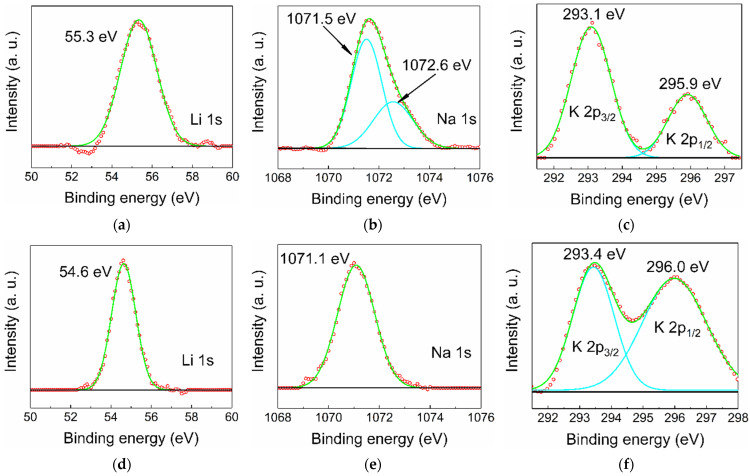
High-resolution XPS spectra of Li, Na, and K dopants in MZ20:Li (**a**), MZ20:Na (**b**), MZ20:K (**c**), MZ30:Li (**d**), MZ30:Na (**e**), and MZ30:K (**f**) alloys. Green lines represent fitted curves, turquoise lines—peak components, red circles—experimental point, black lines—baseline.

**Figure 5 materials-16-05341-f005:**
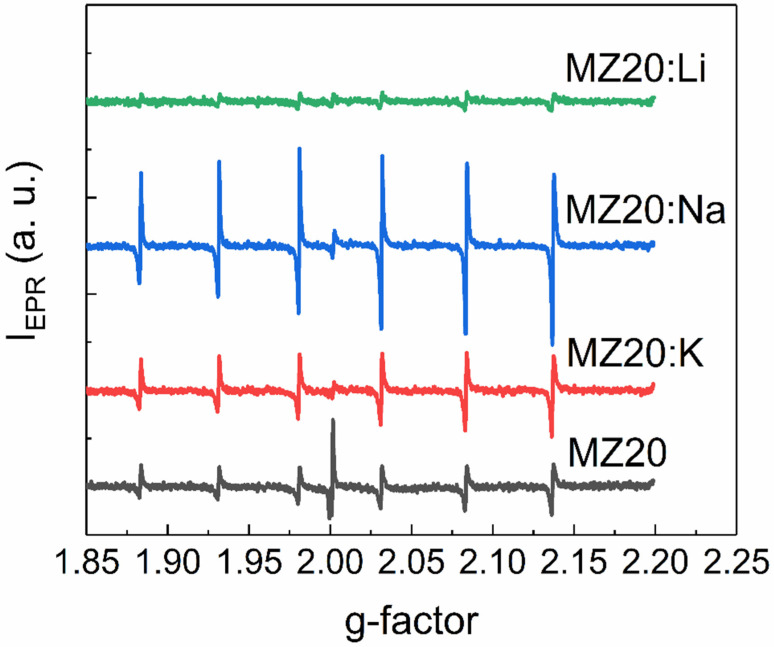
EPR spectra of MZ20:X samples.

**Figure 6 materials-16-05341-f006:**
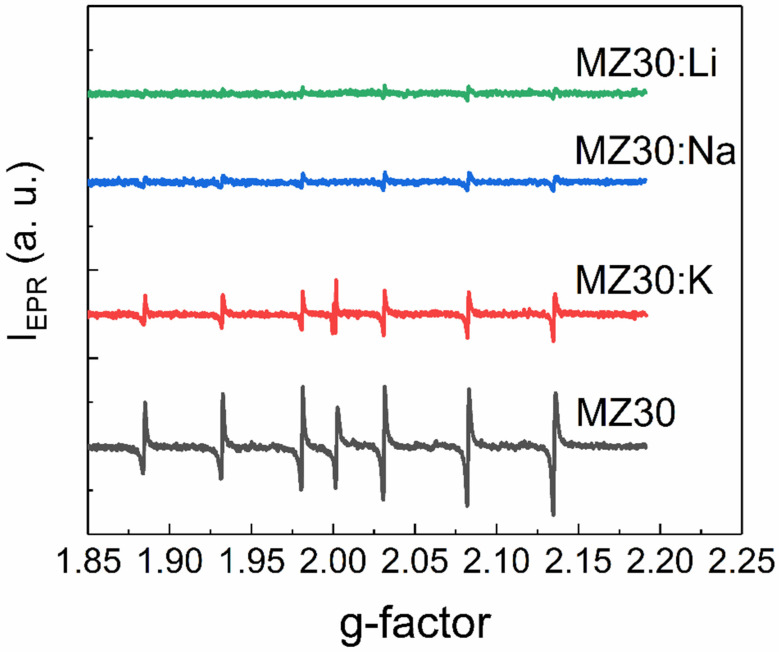
EPR spectra of MZ30:X samples.

**Figure 7 materials-16-05341-f007:**
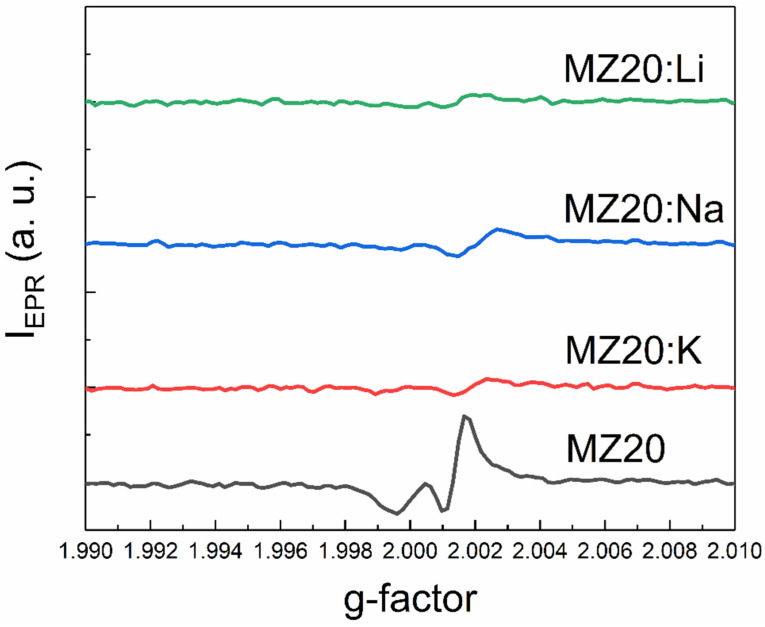
The shape of the EPR line with g ≈ 2 of MZ20:X samples.

**Figure 8 materials-16-05341-f008:**
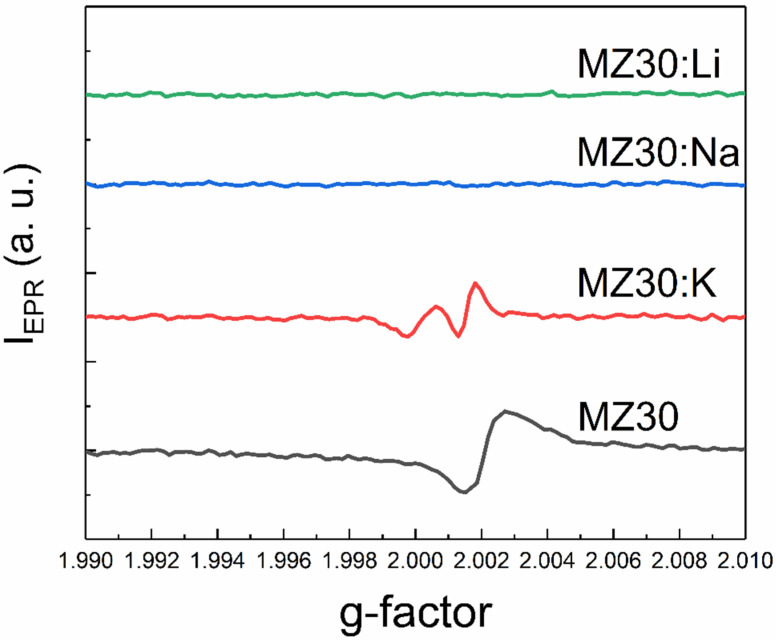
The shape of the EPR line with g ≈ 2 of MZ30:X samples.

**Figure 9 materials-16-05341-f009:**
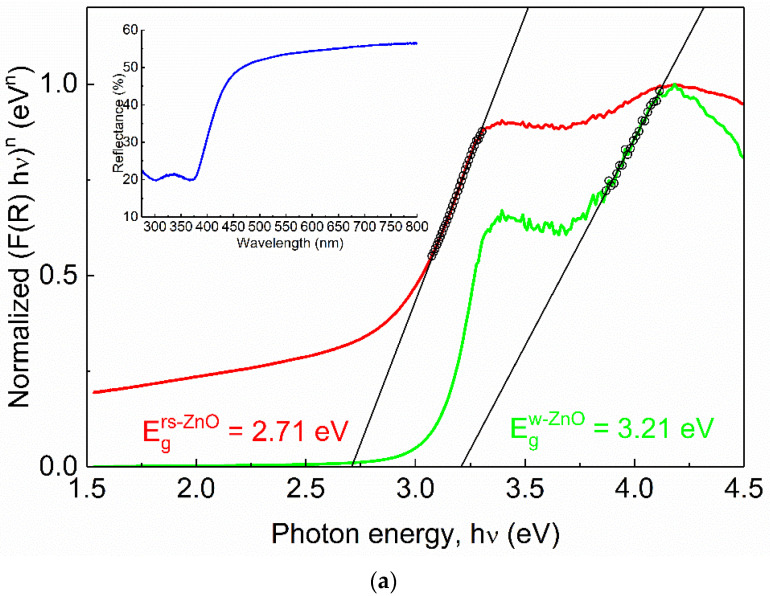
Normalized Tauc plots of rs-ZnO (**a**), MZ20 (**b**) and MZ30 (**c**). Red curves correspond to indirect transitions (the plot constructed using *n* = 2 in Equation (4)), green curves correspond to direct transitions (the plot constructed using *n* = ½ in Equation (4)). Hollow black circles and black lines denote linearized segments of the graphs. Insets show original reflectance spectra. Insets show original reflectance spectra (blue curves).

**Table 1 materials-16-05341-t001:** Mn^2+^ concentration in rs-Mg_x_Zn_1−x_O:X, 10^16^ g^−1^.

Composition	Undoped	K-Doped	Na-Doped	Li-Doped
20% MgO	2.7	10.3	6.3	3.2
30% MgO	5.6	3.1	2.2	0.27

**Table 2 materials-16-05341-t002:** Concentration of paramagnetic centers with g-factor close to two, 10^15^ g^−1^.

Composition	Undoped	K-Doped	Na-Doped	Li-Doped
20% MgO	11	4.0	3.4	3.5
30% MgO	13	4.7	-	-

## Data Availability

The data presented in this study are available on request.

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
