# Peer review of "High-Pressure Synthesis of Cubic ZnO and Its Solid Solutions with MgO Doped with Li, Na, and K"

_materials, 2023, doi:10.3390/ma16155341_

Round 1

Reviewer 1 Report

The current study entitled “High pressure synthesis of cubic ZnO and its solid solutions 2 with MgO doped with Li, Na, and K” looks so interesting for readers and publishing.

However, some comments should be considered :

-          What are the main advantages of the preparation of ZnO under high pressure to the lower preparations? For examples:

Basyooni, M., Shaban, M. & El Sayed, A. Enhanced Gas Sensing Properties of Spin-coated Na-doped ZnO Nanostructured Films. Sci Rep 7, 41716 (2017). https://doi.org/10.1038/srep41716

-          Can you highlight the main research question behind your study?

-          The abstract has to enrich a bit more with your real findings in this study.

-          Can you add the texture coefficient calculation from the above reference to highlight the role of the MgO additive and draw attention to the mechanical properties of ZnO before and after the MgO doping?

-          In Fig 2, it would be better if you have a small scale for the TEM images, around max 100 nm or less, so the readers could imagine the size of the particles. the same is true for S3 in the supporting information section

-          Fig 9 a, c, I think the Eg estimation in the green color curve is not correct. For those, the Eg values should be smaller, around 3, so please recorrect it. 

Moderate editing of English language required

Author Response

Response to Reviewer 1 Comments

The current study entitled “High pressure synthesis of cubic ZnO and its solid solutions 2 with MgO doped with Li, Na, and K” looks so interesting for readers and publishing.

However, some comments should be considered :

Point 1: What are the main advantages of the preparation of ZnO under high pressure to the lower preparations? For examples: Basyooni, M., Shaban, M. & El Sayed, A. Enhanced Gas Sensing Properties of Spin-coated Na-doped ZnO Nanostructured Films. Sci Rep 7, 41716 (2017). https://doi.org/10.1038/srep41716

Response 1: Indeed, there are many facile routes, which can be emloyed for obtaining ZnO in its ground-state hexagonal wurtzite crystal structure. However, in our work, the aim was to obtain not wurtzite ZnO but metastable rock-salt ZnO with the cubic structure that can only form under high pressure. Consequently, here it was not about advantages of the chosen synthetic approach but rather about the necessity. Perhaps, this point was not entirely clear from the introduction. Therefore, we thank you for drawing our attention to this fact and include certain clarifications in lines 48-49.

Point 2: Can you highlight the main research question behind your study?

Response 2: Our main goal was to investigate weather it is possible to obtain rock-salt ZnO phase doped with Li, Na and K as it was theoretically predicted that this could lead to p-type conductivity in ZnO. To our knowledge, there were no previous attempts to synthesise Li, Na or K doped ZnO in the rock-salt structure. We believe that the inclusion we made in lines 56-57 should emphasize the question in study.

Point 3: The abstract has to enrich a bit more with your real findings in this study.

Response 3: The two main findings of this work are: a) it is, in fact, possible to obtain rock-salt ZnO doped with Li, Na and K, for which we provide a synthetic technique, and b) Li, Na and K indeed act as acceptors in rock-salt ZnO in accordance with the theoretical predictions. We hope that this brief summary answers your question. However, these two points are already included in the abstract, therefore, we do not believe that any adjustments need to be made.

Point 4: Can you add the texture coefficient calculation from the above reference to highlight the role of the MgO additive and draw attention to the mechanical properties of ZnO before and after the MgO doping?

Response 4: The texture coefficient could be of importance for oriented structures such as i.e. thin films as for those it can serve as a measure of uniformity. For this reason, if we understood it correctly, the texture coefficient was calculated in the above reference. However, our samples are essentially sintered powders and were studied by X-ray diffraction in the powder form. The influence of texture is minimal in this case, consequently, we do not believe that calculating the texture cofficient would be relevant for our material.

Point 5: In Fig 2, it would be better if you have a small scale for the TEM images, around max 100 nm or less, so the readers could imagine the size of the particles. the same is true for S3 in the supporting information section

Response 5: We have to disagree because many of the particles represented in the TEM images are around 100 nm in diameter, which is essential for nanocrystalline rock-salt ZnO since its stability is related to the particle size as explained in detail in (Baranov, A. N.; Sokolov, P. S.; Tafeenko, V. A.; Lathe, C.; Zubavichus, Y. V.; Veligzhanin, A. A.; Chukichev, M. V.; Solozhenko, V. L. Nanocrystallinity as a Route to Metastable Phases: Rock Salt ZnO. Chem. Mater. 2013, 25, 9, 1775-1782; https://doi.org/10.1021/cm400293j). In our opinion, providing a greater magnification would make it difficult to compare the size of the particles.

Point 6: Fig 9 a, c, I think the Eg estimation in the green color curve is not correct. For those, the Eg values should be smaller, around 3, so please recorrect it.

Response 6: The green color curves are the same reflecance spectra (given in the insets) as the red ones, but reconstructed using n = 1/2 (direct transition) instead of n = 2 (indirect transition) in the Kubelka-Munk equation. If we understand your point correctly, you mean that the value of Eg = 3 should arise from approximating the bottom linear part of the plot to the x axis. However, as we interpreted it, this part of the plot corresponds to the indirect transition in rock-salt ZnO as the dominant phase and is the same as the one linearized in the red curve, only using a different parameter n. Therefore, linearizing this part of the plot in the green curves would be incorect because these curves are given in the coordinates correspondig to n = 2. However the top linear part corresponds to the direct absorption edge that appears due to the presence of either wurtzite ZnO phase or multiple MgO-containg phases. And this linear part is the one approximated in the green curves.

At the same time, we are extremely grateful for your comment as it drew our attention to the fact that in our original draft Fig 9a, and Fig 9c are identical, which was obviously not meant to be the case. We corrected this mistake by inserting the proper graph in Fig 9c.

Reviewer 2 Report

The manuscript reports the high-pressure synthesis and characterization of Li, Na, or K-doped ZnO and MgZnO. The authors clearly justified the research need. The discussion of the results is reasonable, but the following comments should be addressed for more clarification.

1) Figure 1 shows the stable formation of rs phase with a higher MgO amount with the support of XRD data. The reasons for making a stable rs phase need to be discussed. The result section also needs to compare the obtained experimental results with other published articles, for example, the amount of MgO to stabilize rs-ZnO phase, MgO-ZnO phase diagram, etc.

2) XRD results may need labels to show crystallographic planes. Lines 167 mentioned (101) and (111), but it isn't easy to know which peaks correspond to certain crystallographic planes.

3) The authors used equation 2 to estimate crystallite size (35 nm) (Line 185). The authors may mention the value of K shape factor used for the calculation. When comparing the literature, 35 nm crystallite size seems to be a bigger size for the stabilizing rs phase. The authors need to compare the obtained results with reported articles and add the discussion on the effects of crystallite size on the rs phase stabilization.

4) When discussing the types of vacancies (Lines 269-273), Vo+ notation might confuse readers about the valence state of an oxygen vacancy (Is it doubly charged oxygen vacancy Vo++ or singly charged oxygen vacancy Vo+?)

5) To explain the insignificant effect of doping in ZnO-MgO systems, the authors claim the formation of Mg-rich and Mg-poor phases.

In Lines 291-293, the authors claimed that XRD results support the existence of MgO-rich and Mo-poor crystallites. Such explanations needed to be clearly given in the result section of Figure 1. Based on the conclusion section, this reviewer thought that XRD results showed a macroscopically homogenous phase, but other data suspected a microscopically non-homogenous Mg phase.

In Lines 228-232, the authors mentioned a change in the chemical environment could result in uneven distribution of cations. However, the authors did not provide convincing evidence to support the existence of Mg-rich and Mg-poor regions in their samples, which is very important to draw the mechanisms and conclusions.

6) Minor grammatic errors need revision. Line 52: “Recently, has been shown by first-principles calculations…….) need “there”. Line 292: “Mg-rich” instead of “Mg-reach”

English is fine. A minor change of grammatical errors is needed.

Author Response

Response to Reviewer 2 Comments

The manuscript reports the high-pressure synthesis and characterization of Li, Na, or K-doped ZnO and MgZnO. The authors clearly justified the research need. The discussion of the results is reasonable, but the following comments should be addressed for more clarification.

Point 1: Figure 1 shows the stable formation of rs phase with a higher MgO amount with the support of XRD data. The reasons for making a stable rs phase need to be discussed. The result section also needs to compare the obtained experimental results with other published articles, for example, the amount of MgO to stabilize rs-ZnO phase, MgO-ZnO phase diagram, etc.

Response 1: In general, stabilization of rs-ZnO by alloying with MgO (or other oxides with the cubic structure) under high pressure is a well-established fact metioned in the introduction. However, we absolutely agree that more basis should be provided to this statement and for this reason we included two new references in line 49 (No. 21 and 22) to both experimental and theoretical studies that explain the thermodynamic and kinetic reasons of rock-salt phase stabilization and thus prove the validity of our approach.

Point 2: XRD results may need labels to show crystallographic planes. Lines 167 mentioned (101) and (111), but it isn't easy to know which peaks correspond to certain crystallographic planes.

Response 2: This is a good point. We reorganized Fig. 1, Fig. 3 and Fig. S1, accordingly.

Point 3: The authors used equation 2 to estimate crystallite size (35 nm) (Line 185). The authors may mention the value of K shape factor used for the calculation. When comparing the literature, 35 nm crystallite size seems to be a bigger size for the stabilizing rs phase. The authors need to compare the obtained results with reported articles and add the discussion on the effects of crystallite size on the rs phase stabilization.

Response 3: Thank you for noticing that we did not mention the value of K. We used the standrad value of 0.9, we now added this in the text in line 147.

There are two conditions for stabilizing rs-ZnO via nanocrystallinity. The average crystallite size of the precusrsors should not exceed ca. 45 nm. In our case, it was estimated to be 35±5 nm as metioned in line 185. Additionally, there should be no impurity particles of micrometer size present. We did not observe any such particles during TEM scanning in our samples. We understand that it cannot be considered a hard evidence, however, the fact that a nearly pure rock-salt phase was obtained according to XRD seems to suggest that our precursors had sufficiently even particle distributions. Overall, we added some clarification in the text in lines 179-180 and lines 188-191 referring to the original experiment described in (Baranov, A. N.; Sokolov, P. S.; Tafeenko, V. A.; Lathe, C.; Zubavichus, Y. V.; Veligzhanin, A. A.; Chukichev, M. V.; Solozhenko, V. L. Nanocrystallinity as a Route to Metastable Phases: Rock Salt ZnO. Chem. Mater. 2013, 25, 9, 1775-1782; https://doi.org/10.1021/cm400293j).

Point 4: When discussing the types of vacancies (Lines 269-273), Vo+ notation might confuse readers about the valence state of an oxygen vacancy (Is it doubly charged oxygen vacancy Vo++ or singly charged oxygen vacancy Vo+?)

Response 4: We are grateful for this suggestion. We now specified it in lines 269 and 278 where we first use this notation.

Point 5: To explain the insignificant effect of doping in ZnO-MgO systems, the authors claim the formation of Mg-rich and Mg-poor phases.

In Lines 291-293, the authors claimed that XRD results support the existence of MgO-rich and Mo-poor crystallites. Such explanations needed to be clearly given in the result section of Figure 1. Based on the conclusion section, this reviewer thought that XRD results showed a macroscopically homogenous phase, but other data suspected a microscopically non-homogenous Mg phase.

In Lines 228-232, the authors mentioned a change in the chemical environment could result in uneven distribution of cations. However, the authors did not provide convincing evidence to support the existence of Mg-rich and Mg-poor regions in their samples, which is very important to draw the mechanisms and conclusions.

Response 5: We understand your concern and would like to offer the following explanation. First, it is actually impossible to tell from XRD without profile refinement that the system consists of more than one phase (such a work is a currently underway and we hope to publish its results separately in the near future since they involve calculations not particularly suitable for a more “synthetic” work like this one). What was written in lines 291-293 was a mistake, it was meant to say “XPS” instead of “XRD”. We corrected this along with a minor text revision. At the same time, what is written in the conclusion section remains correct.

As for your second point, we believe that there was a misunderstanding because as said in the text, the change in the chemical environment of Li, Na or K is due to uneven distribution of Zn2+ and Mg2+ cations and not vice versa.

Overall, there are several methods namely XPS, EPR and diffuse reflectance spectroscopy that all independently imply that the ZnO-MgO system in the studied MgO concentration range while still macroscopically homogeneous, but microscopically is on the verge of decomposition. Additionally, a similar effect was previously observed in the cognate ZnO-NiO system as mentioned in (Sokolov, P. S.; Baranov, A. N.; Solozhenko, V. L. Phase Stability and Thermal Expansion of ZnO Solid Solutions with 3d Transition Metal Oxides Synthesized at High Pressure. J. Phys. Chem. Solids. 2023, 180, 111437; https://doi.org/10.1016/j.jpcs.2023.111437.). However, we would like to reinstate that within the boundaries of this particular work, the suggestion of Mg-rich and Mg-poor regions existence is only a hypothesis that reasonably explains the experimental data.

Point 6: Minor grammatic errors need revision. Line 52: “Recently, has been shown by first-principles calculations…….) need “there”. Line 292: “Mg-rich” instead of “Mg-reach”

Response 6: Thank you for pointing out these errors. We now made appropriate corrections.

Reviewer 3 Report

This is an interesting report on the high pressure syntheses of pure cubic ZnO and mixed phases with MgO doped with alkali metals. The article is sound and the choice of the experimental methods is correct. I have no major comments, but I would like to place two minor, though still important, suggestions:

- the diffraction patterns in Figure 1 differ quite significantly, also in terms of the background. The pattern of MZ15 is the most noisy one, and while the intensity is arbitrary, I presume that the signal-to-noise ratio is quite poor in this case. It would be good to comment on this discrepancy and include the 2D XRD images before integration as the insets. Otherwise, if the detector was equipped with a 1D detector (this is not clear from the Materials and Methods section if the Rigaku D/Max 2500V/PC comes with an area detector), the differences may be due to the preferred orientation of the crystallites in the sample;

- the TEM diffraction image included in Figure 2(b) indicates a single-crystalline character of the sample after the synthesis. Did the authors attempt to perform a single crystal electron diffraction study on the synthesized sample, which might actually help to resolve its crystal structure and answer the question on its chemical composition?

Author Response

Response to Reviewer 3 Comments

Point 1: The diffraction patterns in Figure 1 differ quite significantly, also in terms of the background. The pattern of MZ15 is the most noisy one, and while the intensity is arbitrary, I presume that the signal-to-noise ratio is quite poor in this case. It would be good to comment on this discrepancy and include the 2D XRD images before integration as the insets. Otherwise, if the detector was equipped with a 1D detector (this is not clear from the Materials and Methods section if the Rigaku D/Max 2500V/PC comes with an area detector), the differences may be due to the preferred orientation of the crystallites in the sample;

Response 1:

As for the quality of the X-ray diffraction pattern of sample MZ15, an explanation is given in the text. See the first paragraph in the Results section. Sample MZ15 partially decomposed after pressure release and contains two phases rs and w, so the question concerning the texture or the quality of diffraction pattern for this sample is no sence.

As for detector we used a 1D scintillation detector and now metioned it in the description of the diffractometer.

Point 2: The TEM diffraction image included in Figure 2(b) indicates a single-crystalline character of the sample after the synthesis. Did the authors attempt to perform a single crystal electron diffraction study on the synthesized sample, which might actually help to resolve its crystal structure and answer the question on its chemical composition?

Response 2: We did attempt to capture an individual particle and record its electron diffraction multiple times. Unfortunately, every such particle would contain several spots on the same concentric ring, which is partly why we concluded that the observed particles are primarily polycrystalline as mentioned in the text, even though it is definitely possible that some of them might be single crystals.

Round 2

Reviewer 1 Report

Thank you for your report 

Moderate editing of English language required

Reviewer 2 Report

The authors have satisfactorily responded to the comments and made the necessary changes to the manuscript.